# Effects of Irrigation Patterns Combining Severe Wilting with Complete or Incomplete Recovery by an Irrigation Control System Based on Photographs of Plants on High-Brix Tomatoes

Fei Zhao [1,2], Hideo Yoshida [1,3], Eiji Goto [1,3] and Shoko Hikosaka [1,3,*]

1 Graduate School of Horticulture, Chiba University, 648 Matsudo, Matsudo 271-8510, Chiba, Japan; zhaofei@xcc.edu.cn (F.Z.); yoshida.hideo@chiba-u.jp (H.Y.); goto@faculty.chiba-u.jp (E.G.)
2 College of Agricultural Science, Xichang University, Xichang 615013, China
3 Research Center for Space Agriculture and Horticulture, Chiba University, 648 Matsudo, Matsudo 271-8510, Chiba, Japan
* Correspondence: s-hikosaka@faculty.chiba-u.jp

**Abstract:** We investigated the effects of irrigation patterns combining severe wilting with complete (S_R) or incomplete recovery (S_IR) on the growth, photosynthesis, fruit quality, and yield using a photograph-based irrigation control system. The study was performed in winter with a single sufficient irrigation treatment as Control, S_R, and S_IR. The daily mean maximum of the wilting ratios (W) in the S_R and S_IR was 15.1% and 15.3%, respectively, when W was set at 14%. S_R had the lowest total irrigation frequency of the three treatments. The accumulated cumulative wilting ratio in S_IR was 1.6 times that in S_R. Under water stress, the net photosynthetic rate decreased (S_IR < S_R), rapidly recovering to 73% and 80% of the maximum values following irrigation, respectively. The total amount of irrigation, the plant growth, and the yield were the highest in the Control and those of S_R and S_IR were comparable. S_IR produced the highest-quality fruit. The recovery level affected the fruit quality when the threshold values were similar; therefore, S_IR is appropriate to produce high-Brix tomatoes in winter. Conclusively, the image-based irrigation system could precisely and reproducibly control the irrigation (the most important parameter affecting the growth, yield, and fruit quality of tomatoes) to improve the fruit quality.

**Keywords:** fruit quality; greenhouse; image-based irrigation system; photosynthetic rate; water stress; wilting ratio

## 1. Introduction

Tomato plants are widely cultivated in protective conditions. Flavorful high-Brix tomatoes have high sugar concentrations (Brix) and have gained popularity worldwide [1–3]. The fruit quality of tomato cultivars is predominantly influenced by genetic factors and growing conditions. Water absorption stress treatments using deficit irrigation can increase the Brix values of greenhouse tomatoes [4–6].

High-Brix tomatoes have higher prices than standard tomatoes, and their production requires more skilled irrigation techniques compared to standard tomatoes. When using deficit irrigation for high-Brix tomato production, growers frequently observe wilting conditions and manage irrigation empirically. Frequent irrigation management used by growers is labor-intensive, time-consuming, and neither systematic nor reproducible. Therefore, a simplified and automated irrigation control system is required to address the problems associated with decreasing and aging agricultural populations. For high-Brix tomato production, it is necessary to automate precise irrigation by monitoring the wilting conditions in real time and implementing appropriate irrigation control systems and patterns.

Several irrigation systems control irrigation patterns for standard tomato production by automatically measuring the substrate water content and plant stress levels. Takayama et al. [7] measured the water stress levels in tomato plants by monitoring their projected leaf area from an image of the canopy captured directly above the plants. Owing to the noncontact, nondestructive, and direct characteristics of the photoimaging method for monitoring wilting conditions, automated irrigation based on digital information from photo images may benefit the stable production of high-Brix tomatoes. However, appropriate water management methods for nutrient solutions have not yet been developed to control the degree of plant wilting or Brix in greenhouse tomatoes.

We developed an irrigation control system based on photographs of the tomato canopy to produce high-Brix fruits and found a significantly negative correlation between the degree of wilting and leaf water potential [8]. This system automatically digitizes the degree of plant wilting using projected leaf areas in real time, and precisely irrigates when the degree of wilting reaches a set point. To control the degree of wilting, we set the maximum degree of wilting as a "threshold value" for the commencement of irrigation and the minimum degree of wilting as a "recovering level" for the termination of irrigation. These two set points affect the irrigation frequency and total amount of the nutrient solution, and the degree of wilting changes between the "threshold value" and the "recovering level". This unique and helpful system can control the balance between yield and Brix value by controlling the degree of wilting.

To optimize irrigation patterns, we investigated the effects of two threshold values on the full or partial recovery of tomatoes [9,10]. In the first experiment, in which the irrigation patterns combined moderate (M) or severe (S; water potential was approximately −1.8 MP and plants survived) wilting with full recovery (R), both the M with R treatment (MR) and S with R treatment (SR) showed similar results for the yield and quality of tomato fruits [9] because of the similar total irrigation amount in both treatments throughout the experiment (the irrigation frequency in SR was lower than in MR; the irrigation amount per time in MR was lower). However, these findings revealed that the total amount of irrigation was more effective than the irrigation frequency, threshold value, or cumulative wilting ratio.

In our subsequent study on partial recovery (PR), in which the amount of irrigation per time was the same in both the moderate wilting with partial recovery (MPR) and severe wilting with partial recovery (SPR) treatments, the irrigation pattern in SPR presented an effective irrigation pattern for improving fruit quality, even during the summer [10]. Accordingly, all water stress treatments significantly decreased the total irrigation amount, growth, and yield, but increased the Brix compared to the Control. However, the threshold values did not affect these parameters during recovery. In addition, the lowest irrigation amount could be applied by SPR, which may be an effective irrigation pattern for high-Brix tomato production, but may inhibit growth and yield.

To improve the growth and yield under water stress, it is necessary to maintain the accumulation of photoassimilates by recovering the net photosynthetic rate (Pn). A study showed that the Pn of tomato leaves decreased under water-deficient conditions; however, Pn recovered to almost its maximum value after complete recovery irrigation [9]. In contrast, Pn did not recover to its maximum value after the incomplete recovery irrigation [10]. Complete-recovery irrigation after severe wilting has been suggested to be appropriate for photoassimilate accumulation.

Based on these findings, we hypothesized that S_IR might increase Brix; however, S_R might alleviate or improve the inhibition of photosynthesis, growth, and yield in high-Brix fruits as a result of Pn maintenance compared to incomplete recovery. To examine these hypotheses and find the optimized irrigation patterns, we investigated the effects of irrigation patterns combining severe wilting with complete or incomplete recovery on the leaf photosynthesis, plant growth, and tomato fruit yield and quality.

## 2. Materials and Methods

### 2.1. Plant Materials and Greenhouse

Two three-week-old tomato seedlings (*Solanum lycopersicum* L., 'Furutika'; Takii & Co., Ltd., Kyoto, Japan) were transplanted into a 1.6 L plastic pot filled with substrate made from coconut fiber (coco wool; Hoags Inc., Tokyo, Japan). Since the coconut fiber itself has no nutrients, the nutrient composition was not affected. It also has a pH between 5.5 and 6.2, making it suitable for most crops. The substrate provides aeration, improving the general structure and facilitating root growth and development. The plants were cultivated in an experimental greenhouse (144 m$^2$), which was a Venlo greenhouse with a steel frame structure. The air temperature was automatically adjusted by automatic roof and side-wall ventilators and heating equipment. This ensured a maximum air temperature of 28 °C during the daytime and a minimum of 15 °C during the nighttime. The distance between the two plastic pots was 20 cm. Fluorine-based and polyolefin films were used as covering materials for the roofs and sides of greenhouses, respectively. The seedlings were irrigated with a one-strength nutrient solution (OAT Agrio Co., Ltd., Tokyo, Japan), which had an electrical conductivity (EC) of 2.7 dS m$^{-1}$ and a pH of 6.8.

During the experimental period (24 November 2021, to 16 February 2022 (85 days)), the average daytime and nighttime air temperatures within the greenhouse were 19.4 $\pm$ 0.2 °C and 15.1 $\pm$ 0.1 °C, respectively, and the relative humidity (RH) ranged between 25% and 55%. The vapor pressure deficit (VPD) ranged between 1.4 and 2.1 kPa. The daily light integral (DLI) ranged between 10 and 17 mol m$^{-2}$ d$^{-1}$ on sunny days and the maximum photosynthetic photon flux density (PPFD) ranged between 200 and 800 µmol m$^{-2}$ s$^{-1}$ at noon.

### 2.2. Treatments

After 59 days of transplantation, when the fourth truss was in anthesis, water stress treatments were initiated. There were three treatments in this experiment: (1) a single sufficient irrigation treatment as a Control, (2) a severe wilting–complete recovery (S_R) treatment, and (3) a severe wilting–incomplete recovery (S_IR) treatment. Twenty tomato fruits were used for each treatment. In the Control, each plant received 30 mL of nutrient solution every 15 min between 06:30 and 16:30.

In S_R and S_IR, an irrigation control system based on photographs of tomato plants was used to monitor the degree of wilting of plants as the wilting ratio (W (%)) at 1 min intervals [9,10]. The system took pictures every min, and then calculated the projected leaf area (PLA) in the pictures separately. The calculation of W was based on changes in the PLA of tomato plants, as described in our previous studies [9,10]. A nutrient solution was supplied when W reached the set value of W (W$_{set}$) as the threshold value. Automatic irrigation was performed from 7:30 to 15:00 using the calculated W. For both S_R and S_IR, the maximum PLA$_{ref}$ was obtained by applying 90 mL of nutrient solution per plant between 06:00 and 07:30. Similar to our previous studies [9,10], as a threshold value for severe wilting, a W$_{set}$ of 14% was used. As W reached 14% from 07:30 to 15:00, the nutrient solution was irrigated to recover the plants without wilting in S_R and to encourage incomplete recovery from W$_{set}$ in S_IR. The irrigation amounts per time were 70 and 30 mL in S_R and S_IR, respectively, resulting in W recoveries of 0–2% and 5–7%, respectively. Every 5 min, irrigation was automatically determined according to the weather conditions that affected W$_{set}$. A nutrient solution of 90 mL was applied to each plant between 15:00 and 15:30 in both treatments to prevent it from wilting before nightfall. In the Control treatment, some water was drained from the pots, whereas in the S_R and S_IR treatments, no water was drained.

### 2.3. Measurement Parameters

#### 2.3.1. Cumulative Wilting Ratio

As in our previous study [9,10], we calculated the cumulative wilting ratio (CWR) by summing $\Delta$W per minute for the 7.5 h between 07:30 and 15:00. If W (t) $\leq$ 4% at minute

't', $\Delta W(t) = 0$; otherwise, $\Delta W(t) = W(t) - 4\%$. We calculated the accumulated CWR for the 85-day experiment.

### 2.3.2. Pn Daily and Potential Values of the Individual Leaves at Different Leaf Layers

Diurnal changes in Pn were determined at 31, 37, and 38 DAT (Control); 29, 30, and 36 DAT (S_R); and 28, 34, and 35 DAT (S_IR) using the transparent cuvette of a portable photosynthesis measurement system (LI-6400XT; LI-COR Inc., Lincoln, NE, USA). The daily maximum, minimum, and mean Pn values were calculated from the values of diurnal changes in Pn. A newly expanding leaf on the sixth leaf from the apex of each treatment was continuously measured from 8:30 to 15:30 on three randomly selected plants from each treatment. Pn was measured under the following environmental conditions: air temperature of 25 °C, RH of $40 \pm 10\%$, $CO_2$ concentration of 400 µmol mol$^{-1}$, and PPFD of 800 µmol m$^{-2}$ s$^{-1}$. The potential value of Pn in the upper, middle, and bottom leaf layers, each containing seven to eight leaves, was measured from 10:00 to 12:00 h at 39 and 78 DAT, respectively, when the RH was $55 \pm 10\%$.

### 2.3.3. Plant Growth

At the end of the experiment (83 d after treatment initiation (DAT), 16 February 2022), the plant length was determined using a tape measure from the bottom to the top of the plant. The stem diameter above each truss was measured using a digital caliper. The total leaf area (including the removed leaves) was determined using a conventional leaf area meter (LI-3000C; LI-COR Inc., Lincoln, NE, USA). The fresh weights (FWs) of each plant organ, including the stem, remaining and removed leaves, removed axillary buds, flowers, fruit trusses, and harvested fruits, were measured using an electronic balance (ASP2102; AS ONE Corp., Osaka, Japan). After one week of drying at 80 °C in a dry oven (MOV-202F(U); PHC Holdings Corp., Tokyo, Japan), the dry weight (DW) was determined. The DW divided by FW was used to calculate the dry matter ratio (%) of each organ. These parameters were measured in three randomly sampled plants from each treatment.

### 2.3.4. Fruit Yield and Quality

To determine the quality and yield of the fruits, the third to eighth fruit trusses, which were subjected to each treatment from anthesis to harvest, were harvested. A total of five harvests (56, 63, 70, 76, and 84 DAT) were conducted during the experiment. From each truss, fruits were counted and weighed. The Brix and acidity were measured nondestructively using a Brix acidity analyzer (Fruit selector, K-BA100R; Kubota Corporation, Osaka, Japan) on six fruits from each truss.

### 2.3.5. Air Temperature Accumulation from Anthesis to Harvest

As described in our previous studies [9,10], the air temperature accumulation (ATC) of the fruits was recorded for 12 fruits from the fifth to sixth fruit trusses. The Control had an ATC of approximately 1172 °C, while both S_R and S_IR had an ATC of approximately 1165 °C. Finally, the anthesis date of the fruits harvested five times using ATC was estimated.

### 2.4. Statistical Analysis

Using one-way analysis of variance (ANOVA), all data, except the accumulated yield and number of harvested fruits, were analyzed with SPSS statistical software (version 24.0; IBM Corp., Armonk, NY, USA). At $p < 0.05$, the Tukey–Kramer test demonstrated significant differences among the Control, S_R, and S_IR.

## 3. Results

### 3.1. Wilting Ratio Transition

In both S_R and S_IR, a diurnal periodic cycle of wilting and recovery occurred from 7:30 to 15:00, and W reached 14% as $W_{set}$ on sunny (Figure 1) and cloudy days (Figure S1).

In S_R, severe wilting and complete recovery cycles occurred approximately every hour, and the number of cycles was 3–6 on sunny days (Figure 1a). In S_IR, a severe wilting and incomplete recovery (approximately 7%, as half of $W_{set}$) cycle occurred approximately every 0.5 h, and the number of cycles was 4–11 on sunny days (Figure 1b). In both S_R and S_IR, there were fewer cycles of wilting and recovery on cloudy days than on sunny days (Figure S1). Therefore, this resulted in a higher daily irrigation frequency in S_IR compared to S_R.

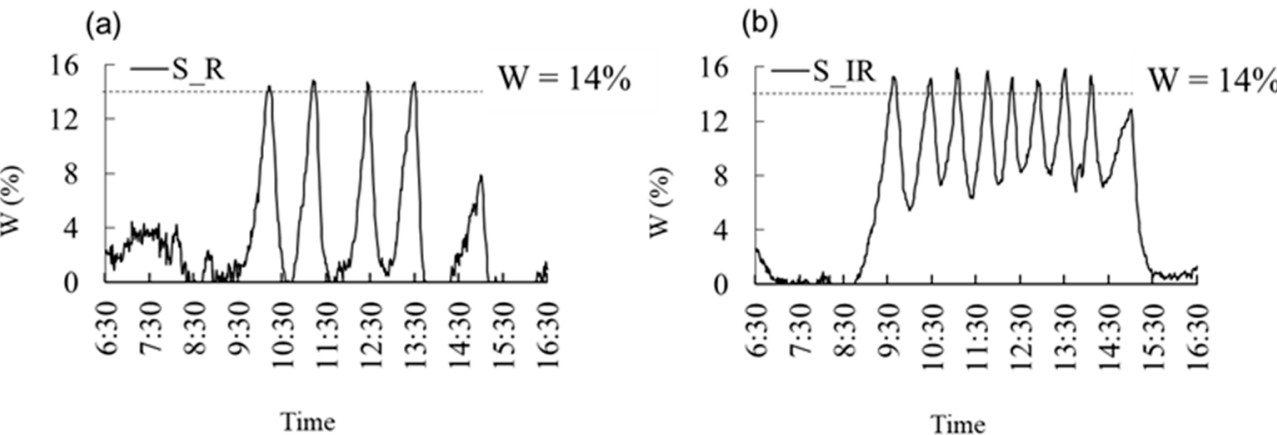

**Figure 1.** Typical changes in wilting ratio (W) in S_R (**a**) and S_IR (**b**) between 06:30 and 16:30 on a sunny day. S_R: severe wilting–complete recovery treatment. S_IR: severe wilting–incomplete recovery treatment. The dotted lines in (**a**) and (**b**) are the set wilting ratios, which are 14%.

In both S_R and S_IR, the W values at 16:30 recovered to 0–2% with sufficient irrigation to avoid wilting at night. During the total 85 days in the experimental period, 61 days (72%) were sunny (the maximum PPFD $\geq$ 600 µmol m$^{-2}$ s$^{-1}$ at noon) and 24 days (28%) were rainy or cloudy. The mean maximum W values for each day in the S_R and S_IR were 15.1% and 15.3%, respectively, when W was set to 14%.

### 3.2. Irrigation Amount, Irrigation Frequency, and Accumulated CWR

The total irrigation amounts and frequencies in the S_R and S_IR groups were substantially lower compared to the Control (Table 1). The total irrigation frequency throughout the experiment in the S_R treatment was the lowest of all treatments. In S_R and S_IR, the total irrigation amounts were 26% (26.6 L) and 23% (23.7 L) of those in the Control, respectively, and the total irrigation amounts were similar in S_R and S_IR. In the S_R and S_IR treatments, the total irrigation frequency was 5% and 9% of what it was in the Control, respectively. At the end of the experiment, the accumulated CWR in S_IR was 1.6 times that in S_R (Figure S2).

**Table 1.** Total amount and frequency of irrigation, and accumulated CWR at the end of the 85-day experiment.

| Treatment | Total Irrigation Amount (L/Plant) | Total Irrigation Frequency (Times) | Accumulated CWR (%) |
|---|---|---|---|
| Control | 102.0 | 3400 | - |
| S_R | 26.6 | 184 | 80,903 |
| S_IR | 23.7 | 302 | 131,594 |

S_R: severe wilting–complete recovery treatment. S_IR: severe wilting–incomplete recovery treatment. CWR: cumulative wilting ratio.

As the same nutrient solution was used in the three treatments, the content of nutrient solutions is proportional to the amount of irrigation. For example, the K$^+$ of the one-strength

nutrient solution was 312 mg $L^{-1}$, then at the end of the 85-day experiment, the $K^+$ values in the Control, S_R, and S_IR were 31.8 g, 8.3 g, and 7.4 g, respectively. Thus, as with $K^+$, the mineral nutrition in S_R and S_IR was 26% and 23% of that in the Control, respectively.

### 3.3. Photosynthesis

The daily maximum, minimum, and mean Pn values of the S_R and S_IR groups were significantly lower than those of the Control (Table 2). No significant differences in the daily maximum and mean Pn were detected between S_R and S_IR, whereas the daily minimum Pn was higher in S_R than that in S_IR. A relatively constant Pn was observed in the Control throughout the day (Figure S3). In contrast, at the S_R and S_IR sites, Pn decreased to minimum values of 62% and 58% of the daily maximum Pn, respectively (Figure S3). Pn recovered rapidly to approximately 80% and 73% of the maximum Pn following irrigation from 15:00 to 15:30 in S_R and S_IR, respectively.

**Table 2.** Effect of irrigation management on the Pn of tomato leaves during the 85-day experimental period.

| Treatment | Daily Max Pn | Daily Min Pn | Daily Mean Pn |
|---|---|---|---|
| Control | 19.5 ± 1.0 a [z] | 15.7 ± 0.6 a | 18.0 ± 0.9 a |
| S_R | 15.4 ± 0.4 b | 9.6 ± 1.0 b | 12.7 ± 0.5 b |
| S_IR | 14.2 ± 1.3 b | 6.2 ± 1.2 c | 10.6 ± 1.2 b |

Pn was determined at 31, 37, and 38 DAT (Control); 29, 30, and 36 DAT (S_R); and 28, 34, and 35 DAT (S_IR). S_R: severe wilting–complete recovery treatment. S_IR: severe wilting–incomplete recovery treatment. The mean ± standard error is used to represent each value (*n* = 3). [z]: the presence of different letters within the same column indicates significant differences among the treatments, as determined by the Tukey–Kramer test at a significance level of 0.05.

Throughout the experiment (39 and 78 DAT), the potential Pn post-irrigation in all treatments was consistently lower in the middle and bottom leaf layers than that in the upper leaf layer (Figure 2). This indicated that the photosynthetic activity of mature or old leaves in the lower leaf layers was lower than that of leaves in the upper leaf layer. During the early experiment period (39 DAT), the potential Pn in the upper leaf layer in both the S_R and S_IR treatments was approximately 75% of that of the Control (Figure 2a), whereas in the later period (78 DAT), the potential Pn in the upper layers in both the S_R and S_IR treatments decreased to approximately 33% of that of the Control (Figure 2b). In addition, the potential Pn values of the middle and bottom leaf layers during the two periods in both the S_R and S_IR treatments were approximately 50% and 30%, respectively, of those in the Control. No significant difference in Pn was detected between S_R and S_IR in each leaf layer throughout the experiment.

### 3.4. Plant Growth

Plant length, leaf area, and fresh and dry weights in the S_R and S_IR groups were similar and lower than those in the Control at 85 DAT (Figure 3). The stem diameters and the number of leaves were also lower in the S_R and S_IR groups than in the Control, whereas the dry matter ratios were higher in the S_R and S_IR groups than in the Control (Table S1). No significant differences in any of the growth parameters were detected between S_R and S_IR.

### 3.5. Yield

The cumulative air temperatures from anthesis to harvest in Control, S_R, and S_IR were 1171.8, 1163.9, and 1165.3 °C, respectively. They were similar due to the similar maturity durations in all trusses in the three treatments; therefore, the anthesis dates of the fruits harvested five times in the three treatments were almost the same. The fresh and dry weights of fruits in the S_R and S_IR treatments were 43.8% and 87.5%, and 35.2% and 83.3%, respectively, of those in the Control at 84 DAT (Table 3). The fresh and dry weights of Control remained relatively constant throughout the experimental period.

However, reductions in fruit fresh and dry weights were observed in the S_R and S_IR groups (Figure 4).

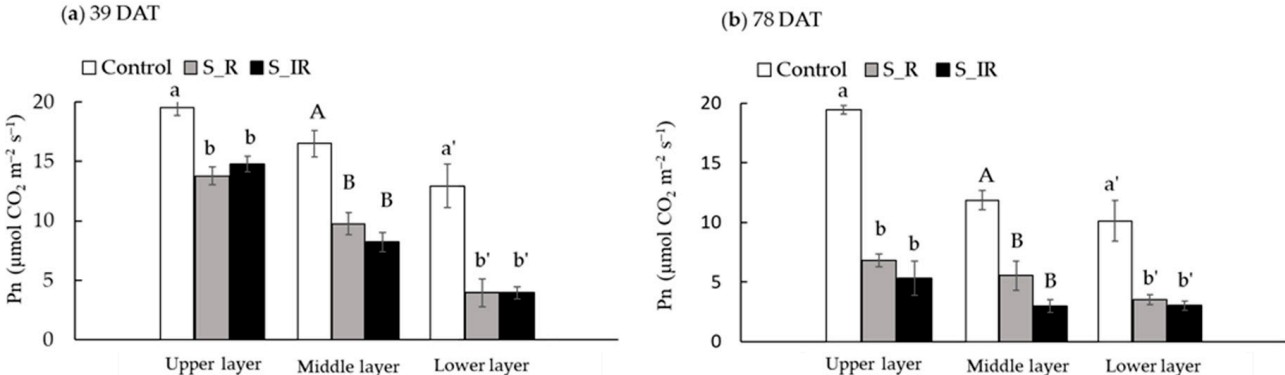

**Figure 2.** Effect of irrigation management on the net photosynthetic rate (Pn) of tomato plants at 39 (**a**) and 78 (**b**) DAT. From 10:00 to 12:00, Pn was measured in each leaf layer. DAT: days after treatment initiation. S_R: severe wilting–complete recovery treatment. S_IR: severe wilting–incomplete recovery treatment. Error bars represent the standard errors of the data (*n* = 3). Different letters indicate significant differences among treatments, as determined by the Tukey–Kramer test at a significance level of 0.05.

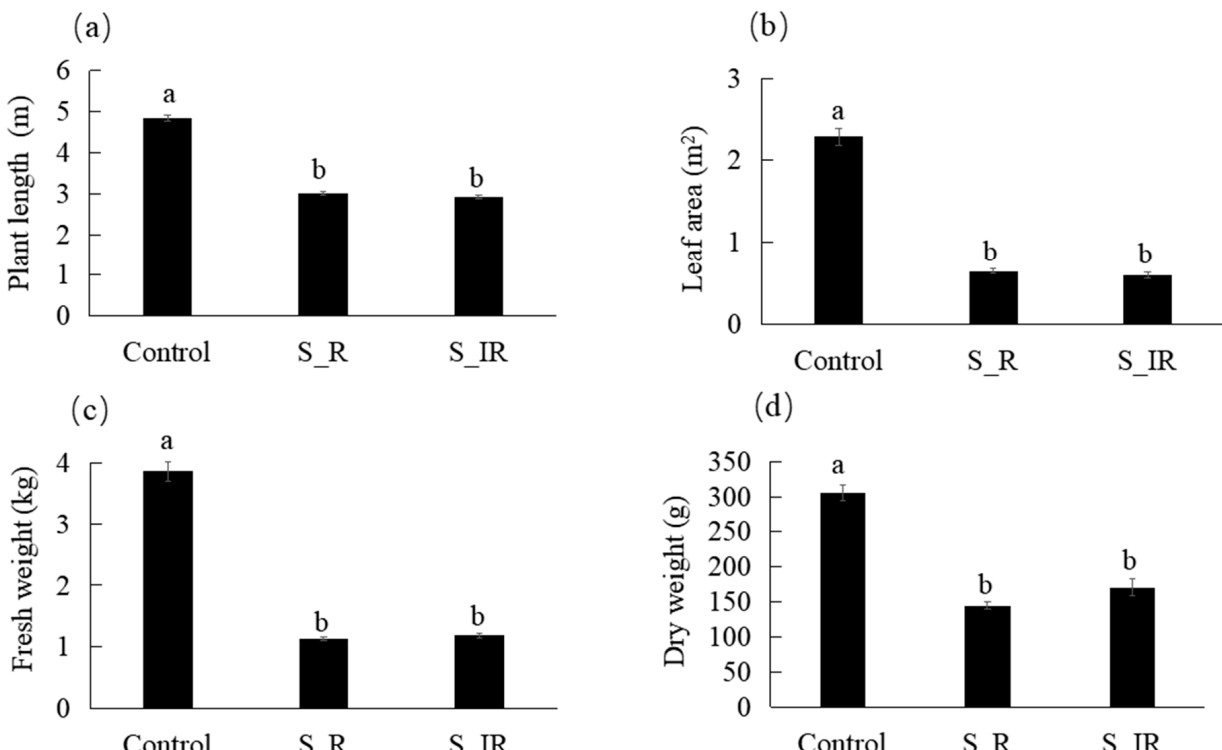

**Figure 3.** Effect of irrigation management on the plant lengths (**a**), leaf area (**b**) and fresh (**c**) and dry (**d**) weights of tomatoes at 85 days after treatment initiation. S_R: severe wilting–complete recovery treatment. S_IR: severe wilting–incomplete recovery treatment. Error bars represent the standard errors of the data (*n* = 3). Different letters indicate significant differences among the treatments in each harvest, as determined by the Tukey–Kramer test at a significance level of 0.05.

**Table 3.** Effect of irrigation management on the fresh and dry fruit weights, accumulated fruit number, and yield of harvested fruit per plant after the 85-day experiment.

| Treatment | Fruit Fresh Weight (g/Fruit) | Fruit Dry Weight (g/Fruit) | Accumulated Number of Harvested Fruits (/Plant) | Accumulated Fruit Yield (kg/Plant) |
|---|---|---|---|---|
| Control | 48.0 ± 1.8 a [z] | 2.4 ± 0.1 a | 11.7 | 0.52 |
| S_R | 21.0 ± 1.2 b | 2.1 ± 0.1 b | 9.0 | 0.21 |
| S_IR | 16.9 ± 0.9 c | 2.0 ± 0.1 b | 9.4 | 0.20 |

S_R: severe wilting–complete recovery treatment. S_IR: severe wilting–incomplete recovery treatment. Twenty tomato fruits were harvested five times (56, 63, 70, 76, and 84 d after treatment initiation (DAT)) in each treatment throughout the experiment. Fruit fresh and dry weights were calculated for the fruits at the fifth harvest. Each value is expressed as the mean plus or minus the standard error (*n* = 4–6). [z]: the presence of different letters within the same column indicates significant differences among the treatments, as determined by the Tukey–Kramer test at a significance level of 0.05. S_R: severe wilting–complete recovery treatment. S_IR: severe wilting–incomplete recovery treatment.

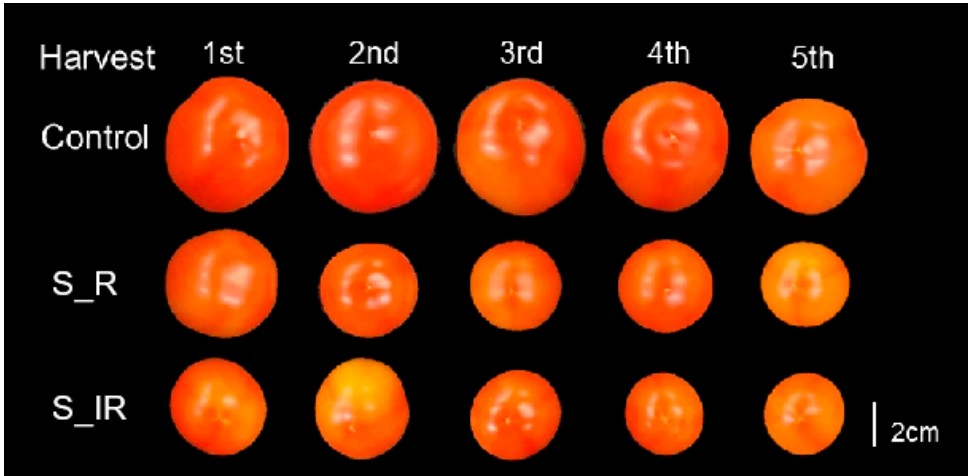

**Figure 4.** Images of fruits in the Control, S_R, and S_IR groups at five-time points (56, 63, 70, 76, and 84 d after the start of treatments (DAT)) throughout the experiment. S_R: severe wilting–complete recovery treatment. S_IR: severe wilting–incomplete recovery treatment. Twenty tomato fruits were harvested in each treatment.

In the Control, S_R, and S_IR treatments, the accumulated fruit yields were 0.52, 0.21, and 0.20 kg per plant, respectively, at 84 DAT, and the ratios in the S_R and S_IR treatments were 40% and 38% of the Control, respectively (Table 3). There was no significant difference in the accumulated number of harvested fruits between the S_R and S_IR treatments.

*3.6. Fruit Quality*

The Brix value and the acidity of the Control remained relatively constant (Table 4). In both the S_R and S_IR treatments, there was a gradual increase in Brix and acidity, reaching maximum values at 76 DAT. In the S_R and S_IR treatments, the Brix and acidity of fruits were higher than those in the Control at each harvest time point, and the values were the highest in the S_IR treatment, with an incomplete recovery level. The Brix values in the S_R and S_IR treatments were 8% higher than those in the Control treatment at 76 DAT (fourth harvest), and 49% and 63% higher at the end of the fifth harvest (84 DAT), respectively. In S_R and S_IR, the acidity was 27% and 39% higher than the Control treatment at 84 DAT, respectively.

**Table 4.** Effect of irrigation management on the Brix and acidity of fruits.

| Treatment | Harvest Time Points (DAT) | | | | |
|---|---|---|---|---|---|
| | **1st (56)** | **2nd (63)** | **3rd (70)** | **4th (76)** | **5th (84)** |
| | Brix (%) | | | | |
| Control | 5.71 ± 0.10 b [z] | 5.53 ± 0.06 c | 5.64 ± 0.09 c | 5.51 ± 0.12 c | 5.63 ± 0.07 c |
| S_R | 7.00 ± 0.08 a | 7.30 ± 0.07 b | 7.90 ± 0.22 b | 8.26 ± 0.19 b | 8.36 ± 0.29 b |
| S_IR | 7.55 ± 0.16 a | 8.24 ± 0.18 a | 9.02 ± 0.23 a | 9.25 ± 0.19 a | 9.19 ± 0.19 a |
| | Acidity | | | | |
| Control | 0.59 ± 0.01 b | 0.58 ± 0.0 b | 0.57 ± 0.01 c | 0.57 ± 0.01 b | 0.58 ± 0.01 c |
| S_R | 0.68 ± 0.01 a | 0.68 ± 0.00 a | 0.71 ± 0.01 b | 0.74 ± 0.02 a | 0.74 ± 0.02 b |
| S_IR | 0.71 ± 0.02 a | 0.76 ± 0.01 a | 0.80 ± 0.01 a | 0.82 ± 0.02 a | 0.80 ± 0.01 a |

Fruits were harvested at five time points (56, 63, 70, 76, and 84 d after treatment initiation (DAT)) throughout the experiment. S_R: severe wilting–complete recovery treatment. S_IR: severe wilting–incomplete recovery treatment. Each value is expressed as the mean plus or minus the standard error (*n* = 4–6). [z]: the presence of different letters within the same column indicates significant differences among the treatments, as determined by the Tukey–Kramer test at a significance level of 0.05.

## 4. Discussion

From the results of wilting ratio, irrigation amount, frequencies, and CWR, the image-based irrigation system using digitized plant wilting conditions in our study could precisely and reproducibly control the irrigation. It seems to contribute to producing high-quality tomatoes without labor and skill.

To determine the proper irrigation management for high-Brix tomato production, we examined the hypothesis that S_IR may increase Brix; however, S_R may also alleviate or improve the inhibition of photosynthesis, growth, and yield in high-Brix fruits compared to S_IR.

### 4.1. Growth and Photosynthesis

Irrigation deficiency inhibits photosynthesis [11–13] and reduces tomato plant growth [14–16]. This results in decreased fresh [17] and dry weights [18] of tomatoes. Consistent with these studies and our previous studies [9,10], this study indicated a marked decrease in the irrigation amount of both S_R and S_IR compared with that of the Control, which decreased photosynthesis and growth. In addition, no significant difference in Pn was detected between S_R and S_IR in each leaf layer throughout the experiment; therefore, this indicates that the total amount of irrigation was the most effective factor for growth and photosynthesis.

Hao et al. [13] reported that the Pn in the upper leaf layer of tomato plants under moderate water stress (soil moisture content of 40–50% of field capacity) decreased to approximately 50% of that under complete irrigation (soil moisture content of 70–80% of field capacity) after 18 days of water stress treatment. In contrast, the Pn in the upper leaf layer in the present study under S_R and S_IR treatments was 75% of that in the Control at 39 DAT (25% reduction). In our previous study [9], no significant differences in Pn were observed in the upper leaf layer between treatments at 30 DAT [9]. The smaller Pn reduction in our study compared with that reported by Hao et al. [13] suggests that a periodic diurnal cycle of wilting recovery in our irrigation control system is beneficial for maintaining the net photosynthetic rate when exposed to conditions of long-term water stress.

The Pn values of S_R and S_IR were not significantly different, and the Pn reduction levels were similar in S_R and S_IR when W reached 14%. Although the percentage of Pn recovery in S_R following irrigation was higher than that in S_IR, this difference did not affect growth throughout the experiment.

In addition, although sufficient irrigation was applied in the morning and late afternoon to prevent wilting, the leaves in S_R and S_IR with small amounts of irrigation tended to be yellow at the end of the experiment. This appeared to be a mineral nutritional deficit, because the concentration of the nutrient solution was constant. Therefore, to avoid

nutrient deficits under water stress, additional mineral nutrients should be supplied to the roots of plants using high-concentration solutions under osmotic stress.

### 4.2. Yield

Consistent with numerous previous studies revealing that water stress reduces fruit yield [14,15,19], there was a decrease in fruit yield (including the number and fresh weight of fruits) in S_R and S_IR groups compared to the Control; however, there were no significant differences in yield in the S_R and S_IR groups. Water stress can reduce photosynthesis in tomato plants [9,11–13] and inhibit the production and transfer of photosynthetic products to the fruits. In our study, a reduction in the total irrigation amount in S_R and S_IR inhibited leaf area expansion and reduced the Pn in each leaf layer, resulting in lower fruit dry weights than that of the Control. Although the recovery level and total amount of irrigation were higher in S_R than in S_IR, there were no significant differences in the fruit yield between S_R and S_IR. This indicated that the slight difference between the two treatments did not result in a significant difference in fruit yield.

The decrease in tomato yield resulting from water stress is primarily attributable to reduced fresh fruit weights [19,20]; therefore, fruit yield declines proportionately to irrigation reductions [10,21,22]. Compared to the Control, total irrigation amounts in the S_R and S_IR treatments in this study were roughly 26% and 23%, respectively, and tomato yields were 40% and 38%, respectively. This was consistent with the results of our previous studies [9,10].

These results demonstrated that the recovery level was not a dominant factor for photosynthesis, growth, or yield, whereas the threshold value strongly affected photosynthetic activity via the total irrigation amount. In addition, the approximate fruit yield and fresh fruit weight could be simulated and controlled based on the total irrigation amount, regardless of the cumulative wilting ratio and irrigation frequency.

### 4.3. Fruit Quality

Fruit yield and quality, including other fruit vegetables, have a trade-off relationship; water stress reduces yield, but frequently improves tomato fruit quality [17,20]. Our results also showed an increase in fruit quality in the S_R and S_IR treatments compared to the Control, corresponding to the total irrigation amount. Machado and Oliveira [19] reported that the Brix of tomato fruits increased with the decrease in irrigation amount. Throughout the experiment, fruit quality gradually improved with a decrease in fruit size, and Brix was >8% after 84 days (in the current study) or 79 days [9] of water stress treatment.

We compared the current study with our previous study [9], which had similar total irrigation amounts for the water stress treatments in each experiment. The full recovery examined in our previous study [9] (MR and SR) showed similar Brix values between treatments; however, in the current study, the Brix value in S_IR was higher than that in S_R. The wilting ratio under full (complete) recovery irrigation conditions in our previous study [9] ranged between 0% and the threshold value, indicating a noncontinuous water stress state. In contrast, the wilting ratio under partial (incomplete) recovery irrigation conditions in the current study was maintained at 50–100% of the threshold value, that is, under a continuous water stress state. In immature fruits, continuous water stress encourages starch accumulation [19] and converts starch into hexose in mature fruits. Therefore, continuous water stress in S_IR may promote starch accumulation in immature fruits, resulting in a higher Brix value than in S_R.

We established that the recovery level and irrigation frequency were different between the S_R and S_IR treatments, but similar total irrigation amounts resulted in a comparable yield and plant growth between the two treatments. These findings suggest that when the threshold value does not exceed the maximum threshold value (14% in this study) at which plants can survive and continuously produce fruit, the total amount of irrigation is the primary factor affecting tomato growth and yield. In addition, the recovery level affected the Brix when the threshold values were high.

In the results of our previous study [9] and the current study, the Brix values of fruits in SR [9] and S_R were >8% at the end of both experiments conducted in winter. In addition, the daily CWR values of SR [9] and S_R were similar. These results demonstrate the reproducibility of high-Brix tomato production using this system. The Brix of fruits in our previous study [10] did not reach 8% after 92 days of water stress treatment in summer, which was lower than the Brix of fruits in winter (9%) in our previous study [9] and the current study. This indicates that excessively high air temperatures (approximately over 30 °C) are not suitable for producing high-Brix tomatoes because of the inhibition of enzyme activity for sugar accumulation.

Although the irrigation patterns of MR, SR, S_R, and S_IR produced tomatoes with high Brix values of >8% in winter in our previous study [9] and in the current study, severe water stress was more favorable than moderate water stress for increasing Brix [10]. Therefore, the S_IR irrigation pattern implemented by this system is appropriate for high-Brix tomato production in winter and for improving Brix production in summer. In terms of the experimental hypothesis, complete recovery irrigation after severe wilting (S_R) did not improve the accumulation of photoassimilates compared with S_IR.

Although further experiments are required to optimize the mineral nutrient application under limited irrigation conditions, our findings provide insights into the technical aspects of high-quality tomato production in a greenhouse. Furthermore, although this study is based on a single cultivar and specific weather conditions, the findings can be applied to other crops and weather conditions with some set-point modifications.

## 5. Conclusions

Here, we revealed that the net photosynthetic rate, plant growth, and yield under irrigation patterns combined with severe wilting and incomplete or complete recovery were similar because of the similar total irrigation amounts in the two irrigation patterns. When the threshold value did not exceed the maximum value of 14%, at which the plants could survive and continuously produce fruit, the total amount of irrigation was the primary factor affecting the growth, yield, and Brix of tomatoes. The recovery level affected the fruit quality when the threshold values were the same. Furthermore, we demonstrated that the image-based irrigation system based on digitized plant wilting conditions could precisely and reproducibly control the irrigation to produce high-quality tomatoes without heavy labor and specialized skills. An irrigation pattern combining severe wilting with incomplete recovery implemented by this system is appropriate for high-Brix tomato production in winter and for improving fruit quality in summer.

**Supplementary Materials:** The following supporting information can be downloaded at: https://www.mdpi.com/article/10.3390/horticulturae9101143/s1, Table S1. Effect of irrigation management on the number of leaves, stem diameter, and dry matter ratios of tomatoes measured at 85 days after the start of treatments. Figure S1. Typical changes in wilting ratio (W) in S_R (a) and S_IR (b) between 06:30 and 16:30 on a cloudy day. S_R: severe wilting–complete recovery treatment. S_IR: severe wilting–incomplete recovery treatment. The dotted lines in (a) and (b) are the set wilting ratios, which are 14%. Figure S2. Accumulated cumulative wilting ratios (CWR) during the 85-day experimental period. The cumulative wilting ratio (CWR) by summating W (>4%) per minute for the 7.5 h between 07:30 and 15:00. S_R: severe wilting–complete recovery treatment. S_IR: severe wilting–incomplete recovery treatment. DAT: days after treatment initiation. Figure S3. Typical diurnal changes in the net photosynthetic rate (Pn) and wilting ratio (W) in the Control (a), S_R (b), and S_IR (c) from 08:30 to 15:30 on a sunny day. S_R: severe wilting–complete recovery treatment. S_IR: severe wilting–incomplete recovery treatment. DAT: days after treatment initiation.

**Author Contributions:** F.Z. and S.H. designed the experiments; investigation and data curation were conducted by F.Z.; the analysis and writing of the original draft were performed by F.Z. and S.H.; review and editing of the manuscript were performed by E.G., S.H. and H.Y. All authors have read and agreed to the published version of the manuscript.

**Funding:** This research was funded by Kubota Corporation, Osaka, Japan.

**Data Availability Statement:** Data are contained within the article and Supplementary Materials.

**Acknowledgments:** For technical assistance with the irrigation control system based on images of tomato plants, the authors thank Hironobu Koga, Asako Kataoka, and Masahiko Yasui from the Technology Innovation Research and Development Unit of Kubota Corporation.

**Conflicts of Interest:** The authors declare no conflict of interest. The funders had no role in the study design, collection, analyses, or interpretation of data; in the writing of the manuscript; or in the decision to publish the results.

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
