# Peer review of "Effects of Irrigation Patterns Combining Severe Wilting with Complete or Incomplete Recovery by an Irrigation Control System Based on Photographs of Plants on High-Brix Tomatoes"

_horticulturae, doi:10.3390/horticulturae9101143_

Round 1
Reviewer 1 Report
The present work using a correct experimental and analytical approach analyses the effects of an “Automatic Image-Based Irrigation System” on yield , photosynthetic activity, growth, and quality (Brix) of tomato. The irrigation control system has been developed and extensively investigated by the same authors and the results obtained largely follow what has already been observed in previous works by the same and by other authors. Although there isn't a significant leap in findings compared to prior works, I believe the manuscript warrants publication due to its practical insights into optimizing tomato irrigation methods.
Author Response
Response to Reviewer 1 Comments
Dear Reviewer,
We would like to thank the editor and reviewers for their comments and suggestions. We have revised the manuscript as suggested by the reviewers, and the amendments have been depicted using the " red text " in the revised manuscript.
Below, we have provided our responses to each of your comments and have made the corresponding revisions in the revised manuscript; in this document, our responses and revisions are indicated using the red font. The line numbers in our responses have been included in the revised manuscript.
Reviewer 1’s comments
The present work using a correct experimental and analytical approach analyses the effects of an “Automatic Image-Based Irrigation System” on yield, photosynthetic activity, growth, and quality (Brix) of tomato. The irrigation control system has been developed and extensively investigated by the same authors and the results obtained largely follow what has already been observed in previous works by the same and by other authors. Although there isn't a significant leap in findings compared to prior works, I believe the manuscript warrants publication due to its practical insights into optimizing tomato irrigation methods.
Reply: We would like to express our appreciation for your time and valuable comments on our article.
Thank you for taking the time to read our responses.
Sincerely yours,
Authors

Reviewer 2 Report
This study investigated the effects of irrigation patterns combining severe wilting with complete or incomplete recovery on growth, photosynthesis, fruit quality, and yield using an irrigation control system based on photographs. This study has some practical implications, however, some clarification is needed on the following comments:
1. The abstract does not highlight the research innovation and research significance.
2. Introduction to the physical and chemical properties of the substrate in Plant materials and greenhouse.
3. Please introduce the process of taking photos and how to monitor the degree of wilting of tomato plants based on photos.
4. This paper mainly studies the quality of tomatoes, and whether other qualities other than sugar and acid are monitored.
5. In the conclusion part, we should point out the shortcomings of the thesis and show the limitations of the thesis.
6. Pay attention to the chart format, such as the clarity of the picture.

minor editing is needed.
Author Response
Response to Reviewer 2 Comments
Dear Reviewer,
We would like to thank the editor and reviewers for their comments and suggestions. We have revised the manuscript as suggested by the reviewers, and the amendments have been depicted using the " red text " in the revised manuscript.
Below, we have provided our responses to each of your comments and have made the corresponding revisions in the revised manuscript; in this document, our responses and revisions are indicated using the red font. The line numbers in our responses have been included in the revised manuscript.
Reviewer 2’s comments
This study investigated the effects of irrigation patterns combining severe wilting with complete or incomplete recovery on growth, photosynthesis, fruit quality, and yield using an irrigation control system based on photographs. This study has some practical implications, however, some clarification is needed on the following comments:
Point 1:
The abstract does not highlight the research innovation and research significance.
Response 1:
Thank you for the suggestion. According to the comment, we have added the research innovation and research significance in Abstract:
L22-24 Conclusively, the image-based irrigation system could precisely and reproducibly control the irrigation (the most important parameter affecting the growth, yield, and fruit quality of tomatoes) to improve fruit quality.
Point 2:
Introduction to the physical and chemical properties of the substrate in Plant materials and greenhouse.
Response 2:
Thank you for pointing this out. Accordingly, we have added the content to introduce the physical and chemical properties of the substrate in plant materials and greenhouse as follows:
L100-107 Since the coconut fiber itself has no nutrients, the nutrient composition was not affected. It also has a pH between 5,5 and 6,2, making it suitable for most crops. The substrate pro-vides aeration, improving the general structure and facilitating root growth and development. The plant were cultivated in an experimental greenhouse (144 m2); a Venlo green-house with a steel frame structure. Air temperature was automatically adjusted by automatic roof and side-wall ventilators and heating equipment. This ensured a maximum air temperature of 28 °C during the daytime and a minimum of 15 °C during the nighttime.
Point 3:
Please introduce the process of taking photos and how to monitor the degree of wilting of tomato plants based on photos.
Response 3:
Following your comment, we have modified and added the sentences in the revised manuscript as follows:
L128-129 The system took pictures every min, and then calculated the projected leaf area in the pictures separately.
Point 4:
This paper mainly studies the quality of tomatoes, and whether other qualities other than sugar and acid are monitored.
Response 4:
Thank you for pointing this out. In this study, we measured sugar and acid as the most important qualities of tomatoes; however, we did not measure other qualities. The figure shows clearly how clean the exterior is. There is room for future research to consider other functional ingredients and scents, but here we first prioritized automatically producing high-sugar tomatoes.
Point 5:
In the conclusion part, we should point out the shortcomings of the thesis and show the limitations of the thesis.
Response 5:
Thank you for the suggestion. We have added the content to clarify the limitations of our study as follows.
L446-451 Although further experiments are required to optimize the mineral nutrient application under the limited irrigation conditions, our findings provide insights into the technical aspects of high-quality tomato production in a greenhouse. Furthermore, although this study is based on a single cultivar and specific weather conditions, the findings can be applied to other crops and weather conditions with some set point modifications.
Point 6:
Pay attention to the chart format, such as the clarity of the picture.
Response 6:
Thank you very much. We increased the resolution of the image according to your comment.
L263-275 shows the new Figure 2 and Figure 3.
We look forward to hearing from you and would be happy to make further changes, if required.
Sincerely yours,
Authors

Reviewer 3 Report
Review of the article " Effects of Irrigation Patterns Combining Severe Wilting with Complete or Incomplete Recovery by an Irrigation Control System based on Photographs of Plants on High-Brix Tomatoes" in the journal "Horticulture".
The research topic is relevant for selection of tomato growing conditions in greenhouse farms
Literature review is not sufficient, it is necessary to expand the list of used literature.
The experimental part is performed and analyzed satisfactorily.
There are the following remarks:
1. In the introduction and in the text of the whole article, more authors pay a lot of attention to the discussion of the results of their two previous works (references 9 and 10), while references to the works of other authors are clearly insufficient
2. The purpose of the paper should be clearly stated.
3. In the Materials and Methods section it is necessary to:
- line 140 - add a reference to the methodology for determining CWR;
- line 155 - specify the date;
- line 161 - specify the brand of the electronic scale;
- line 162 - specify the brand of the desiccator;
- lines 143-176 - specify the specific conditions under which the experiment was conducted;4. It is recommended to systematize the order of material in the Discussion section in accordance with the order of presentation in the Results section.
4. Comments on the Results section.
- Section 3.1 discusses the meteorological conditions of the plant growth period. what significance does this have if the experimental plants were grown under shelter?
- the conclusion (lines 236-237) does not follow from the results and should be clearly justified.
- line 268 - data should be cited.
5. The discussion section discusses mineral nutrition, but there is no information about this in the experimental section. Generally the discussion section gives the impression of lacking a close connection to the experiment conducted
The statistical analysis is convincing.
Once the shortcomings are corrected, the article will meet the rating of the journal Horticulture and after a second review can be recommended for publication.
Author Response
Response to Reviewer 3 Comments
Dear Reviewer,
We would like to thank the editor and reviewers for their comments and suggestions. We have revised the manuscript as suggested by the reviewers, and the amendments have been depicted using the " red text " in the revised manuscript.
Below, we have provided our responses to each of your comments and have made the corresponding revisions in the revised manuscript; in this document, our responses and revisions are indicated using the red font. The line numbers in our responses have been included in the revised manuscript.
Reviewer 3’s comments
Review of the article " Effects of Irrigation Patterns Combining Severe Wilting with Complete or Incomplete Recovery by an Irrigation Control System based on Photographs of Plants on High-Brix Tomatoes" in the journal "Horticulture".
The research topic is relevant for selection of tomato growing conditions in greenhouse farms
Literature review is not sufficient, it is necessary to expand the list of used literature.
The experimental part is performed and analyzed satisfactorily.
There are the following remarks:
Point 1:
In the introduction and in the text of the whole article, more authors pay a lot of attention to the discussion of the results of their two previous works (references 9 and 10), while references to the works of other authors are clearly insufficient.
Response 1:
Thank you for pointing this out. According to the comment, we have added the content as below.
L340 This results in decreased fresh [17] and dry weights [18] of tomatoes.
L394-395 Machado and Oliveira [19] reported that the Brix of tomato fruits increased with the decrease of irrigation amount.
We have also added two references, numbered 17-18.
Point 2:
The purpose of the paper should be clearly stated.
Response 2:
Thank you for pointing this out. We have added the relevant content as below:
L92-95 To examine these hypotheses and find the optimized irrigation patterns, we investigated the effects of irrigation patterns combining severe wilting with complete or incomplete recovery on leaf photosynthesis, plant growth, and tomato fruit yield and quality.
Point 3:
In the Materials and Methods section it is necessary to:
- line 140 - add a reference to the methodology for determining CWR;
Response:
Thank you for your comment. We have added the content as below:
L146-149 As in our previous study [9,10], we calculated the cumulative wilting ratio (CWR) by summating ΔW per minute for the 7.5 hours between 07:30 and 15:00. If W (t) ≤ 4% at minute ‘t’, ∆W(t) = 0; otherwise, ∆W(t) = W(t)−4%. We calculated the accumulated CWR of the 85 d experiment.
- line 155 - specify the date;
Response:
Thank you very much. According to the comment, we have added the information of instruments as below.
L163 At the end of the experiment (85 d after treatment initiation [DAT], February 16, 2022),
- line 161 - specify the brand of the electronic scale;
- line 162 - specify the brand of the desiccator;
Response:
Thank you for pointing this out. Accordingly, we have added the information of instruments:
L169-171 ...... an electronic balance (ASP2102; AS ONE Corp., Osaka, Japan). After one week of drying at 80 °C in a dry oven (MOV-202F(U); PHC Holdings Corp., Tokyo, Japan) ...
- lines 143-176 - specify the specific conditions under which the experiment was conducted;
Response:
Thank you very much. The conditions under which the experiment is performed are described in detail in lines 112 – 118.
Point 4: It is recommended to systematize the order of material in the Discussion section in accordance with the order of presentation in the Results section.
Response:
Thank you for pointing this out. We considered and decided not to change the order to avoid section fragmentation and discuss from several results.
Point 5:
Comments on the Results section.
- Section 3.1 discusses the meteorological conditions of the plant growth period. what significance does this have if the experimental plants were grown under shelter?
Response:
Thank you very much for your comments. High-quality tomatoes can only be produced under facility cultivation conditions. High-quality tomato production requires appropriate drought stress; however, the meteorological conditions will affect the environment inside the facility and thus affect the degree of water stress the tomato growth is subjected to, that is, the wilting ratio used in this paper. Therefore, it is necessary to discuss the meteorological conditions of the plant growth period.
- the conclusion (lines 236-237) does not follow from the results and should be clearly justified.
Response:
Lines 236-237 are a part of the results and not conclusion, thus we were unable to determine which part you mean. However, the results of lines 193-233 were used in lines 334 to 337 in the revised discussion to clarify the conclusion.
- line 268 - data should be cited.
Response:
Thank you for your comments. According to the comment, we have added the data as below.
L284-287 The cumulative air temperatures from anthesis to harvest in Control, S_R and S_IR were 1171.8, 1163.9, and 11165.3, respectively. They were similar due to the similar maturity durations in all trusses in the three treatments; therefore......
Point 6:
The discussion section discusses mineral nutrition, but there is no information about this in the experimental section. Generally the discussion section gives the impression of lacking a close connection to the experiment conducted
The statistical analysis is convincing
Response 6:
Thank you for bringing this to our attention. We have added the content accordingly:
L228-233 As the same nutrient solution was used in the three treatments, the content of nutrient solutions is proportional to the amount of irrigation. For example, the K+ of the one-strength nutrient solution was 312 mg L–1, then at the end of the 85 day experiment, the K+ in the Control, S_R, and S_IR were 31.8 g, 8.3 g, and 7.4 g, respectively. Thus, as with K+, the mineral nutrition in the S_R and S_IR were 26% and 23% of those in the Control, respectively.
Thank you for taking the time to read our responses. We look forward to hearing from you and would be happy to make further changes, if required.
Sincerely yours,
Authors

Round 2
Reviewer 3 Report
I have studied the article with the changes. In my opinion, the article is sufficiently improved and can be recommended for publication.
I have studied the article with the changes. In my opinion, the article is sufficiently improved and can be recommended for publication.